# An integrative network-based approach for drug target indication expansion

**Yingnan Han, Clarence Wang, Katherine Klinger, Deepak K. Rajpal\*, Cheng Zhu \***

Translational Sciences, Sanofi, Framingham, Massachusetts, United States of America

\* Cheng.Zhu@sanofi.com (CZ); Deepak.Rajpal@sanofi.com (DKR)

## Abstract

### Background

The identification of a target-indication pair is regarded as the first step in a traditional drug discovery and development process. Significant investment and attrition occur during discovery and development before a molecule is shown to be safe and efficacious for the selected indication and becomes an approved drug. Many drug targets are functionally pleiotropic and might be good targets for multiple indications. Methodologies that leverage years of scientific contributions on drug targets to allow systematic evaluation of other indication opportunities are critical for both patients and drug discovery and development scientists.

### Methods

We introduced a network-based approach to systematically screen and prioritize disease indications for drug targets. The approach fundamentally integrates disease genomics data and protein interaction network. Further, the methodology allows for indication identification by leveraging state-of-art network algorithms to generate and compare the target and disease subnetworks.

### Results

We first evaluated the performance of our method on recovering FDA approved indications for 15 randomly selected drug targets. The results showed superior performance when compared with other state-of-art approaches. Using this approach, we predicted novel indications supported by literature evidence for several highly pursued drug targets such as IL12/IL23 combination.

### Conclusions

Our results demonstrated a potential global approach for indication expansion strategies. The proposed methodology enables rapid and systematic evaluation of both individual and combined drug targets for novel indications. Additionally, this approach provides novel insights on expanding the role of genes and pathways for developing therapeutic intervention strategies.

**Data Availability Statement:** All relevant data are within the paper and its Supporting information files.

**Funding:** I confirm the commercial affiliation to Sanofi provided support in the form of salaries for

authors YNH, CW, KK, DR, CZ, but did not have any additional role in the study design, data collection and analysis, decision to publish, or preparation of the manuscript. The specific roles of these authors are articulated in the 'author contributions' section.

**Competing interests:** I confirm commercial affiliation to Sanofi does not alter our adherence to PLOS ONE policies on sharing data and materials.

## Introduction

Identification of high-quality drug targets is at the heart of successful drug discovery and development but remains extremely challenging. The early "one gene, one drug, one disease" paradigm [1] has evolved to consider the cellular and physiological context of the target. This opens the potential for a drug target to be a potentially good target for multiple disease indications beyond the first or best established one. Efficient methods for indication expansion are required to realize this potential.

The identification of the right target-indication pair is highly challenging through traditional ad-hoc approaches. The generation of rich and heterogeneous multi-omics data resources, and disease to gene association knowledge recently has enabled the development of several computational approaches intended to systematically expand the therapeutic base of a given target. Among the proposed methods, some of them are similarity-based approaches. For instance, Gottlieb *et al.*, developed a drug indication (PREDICT) method that can identify drug-disease associations and predict new drug indications based on the observation that similar drugs are indicated for similar diseases [2]. Zhao *et al.* established comCIPHER which leveraged Bayesian partition method to identify drug–gene–disease co-modules to relate drug to diseases for novel indication [3]. However, such approaches usually required comprehensive drug and disease information such as chemical structure and treatment profiles. Some of the methods are machine learning based approaches [4–6]. However, the machine learning approaches mainly consider the known drug to disease associations as positive or negative samples, while drug target and disease molecular basis, such as cellular and physiological context, have been largely overlooked.

On the other hand, network-based methods have become popular approaches as they provide a systems-level view of the corresponding biological systems. Many biomedical networks were proven comparable to social or web networks which have scale-free and small-world properties [7], and therefore algorithms developed for analyzing social or web networks can be equally applicable to biological network tasks [8–10].

Since retrieving useful patterns in the networks may uncover the underlying biomedical characteristics and provide possible solutions for medical treatment, several network-based methods have been proposed to discover new indications for approved drugs. The proposed network methods exploit their own generated biomedical networks based on drug and disease profiles or integrate the profiles with a molecular network. For instance, Liu *et al.* integrated a drug-drug networks with a disease-disease similarity network, then applied a random walk algorithm on the network to predict new indications for approved drugs [11]. Chiang and Butte [12] introduced a network-based, guilt-by-association method to expand novel drug indications based on shared treatment profile from disease pairs. Emig *et al.* [13] proposed a network method which took disease signatures as input to generate target profiles, then applied a logistic regression model to train and prioritize targets for drug repositioning. However, the methods were mostly based on establishing a disease phenotype or drug similarity network to identify novel drug indications, there are still some limitations that need to be addressed: For example, as the drug target perturbation is usually accompanied by disruption of its involved biological processes, it is critical to investigate how the drug target and disease's biological processes or pathways are related. Additionally, human genetic evidence on a drug target has been reported to play an increasingly important role on achieving a successful clinical trial [14]. Therefore, it is of great importance to explore how drug targets can achieve therapeutic benefit for different indications through their biological network connectivity while supported by the genetic evidences.

Here, we introduce a network-based approach by integrating genetic evidence with a protein to protein interaction (PPI) network to systematically screen and prioritize disease indications for drug targets. Our approach was inspired by the following scientific rationales: First, validated targets for well-studied indications are usually supported by human genetic evidence. Recent work by Nelson *et al.* [15] showed that among the portfolios of drug targets, the numbers of those with direct human genetic support increased significantly across the drug development pipelines and they can achieve 2-fold higher probability of success. On the other hand, many complex human diseases are genetically associated with multiple variants identified from genome-wide association studies (GWAS) [16]. Those disease associated genes also tend to have a high propensity to interact with each other, forming disease modules or subnetworks in a molecular interaction network. For instance, many immune-mediated diseases reflect perturbation of genes that interact in complex networks [17]. Most importantly, many studies have demonstrated that genetically associated disease genes and validated targets tend to be neighbors in molecular interaction networks [18, 19].

Based on these findings, we hypothesized that for any disease indication, when genetically associated genes are closely connected with the drug target(s) and their neighboring genes in a PPI network, then they could be considered as potential candidates for the drug target. We therefore formulate the target indication expansion problem as prioritizing diseases for drug target(s) through network comparisons. A target subnetwork was defined as a subnetwork that connects the drug target(s) and their neighboring nodes in PPI network, which represents the affected processes or signaling pathways under target perturbation. A disease subnetwork was defined as a subnetwork which contains all (or the majority) of the disease genetically associated genes, with additional essential genes that can form a network module in PPI network.

We present the key steps of our approach in Fig 1: A) For a given drug target, we use a state-of-art node prioritization algorithm (see Materials and methods) to prioritize the nodes that have high connectivity with the drug target in the PPI network. Those top ranked nodes and their direct interactomes were selected to generate the target subnetwork. For each disease, we performed module detection algorithms from disease associated genes to detect the disease subnetwork. B) The network node enrichment between target subnetwork and each disease subnetwork was computed by hypergeometric test to derive the enrichment score. C) Finally, all the diseases were ranked by the enrichment score to identify the top potential indications for the target.

## Results

### Performance evaluation

In order to evaluate our network method performance doesn't occur by chance, we randomly selected 15 drug targets with their FDA approved indications and assessed how well their approved indications can be successfully predicted when compared with the prediction results on random drug targets. The target list was retrieved from Clarivate's MetaCore (https://portal.genego.com), in which each target has at least one FDA approved indication. For all the diseases or indications used in performance evaluation, we took disease collections from DisGeNET [20]. The approved indications from MetaCore were mapped to DisGeNET disease terms for performance evaluation. Table 1 shows the selected drug targets and their corresponding number of approved indications, the full list of all approved indications for the 15 drug targets can be found in S1 File.

For each selected drug target, we randomly selected 100 druggable targets and used the same network approach to compute their average performance. We calculated the sensitivity and specificity for each threshold. Sensitivity refers to the percentage of the associations whose

**Target and Disease Subnetwork Generation**

**Fig 1. Workflow of network-based approach for indication expansion.**

**Table 1. Summarization of drug targets and their number of approved indications in MetaCore.**

| Drug Target | Number of Approved Indications |
|---|---|
| TNF-alpha | 10 |
| Alpha-2A adrenergic receptor | 12 |
| COX-2 (PTGS2) | 20 |
| GABA-A receptor | 6 |
| Histamine H1 receptor | 26 |
| HTR2A | 13 |
| SERT | 11 |
| ACM3 | 20 |
| IFNAR1 | 3 |
| BTK | 2 |
| HTR6 | 7 |
| MCR | 9 |
| NET | 11 |
| Beta-1 adrenergic receptor | 12 |
| Dopamine D2 receptor | 21 |

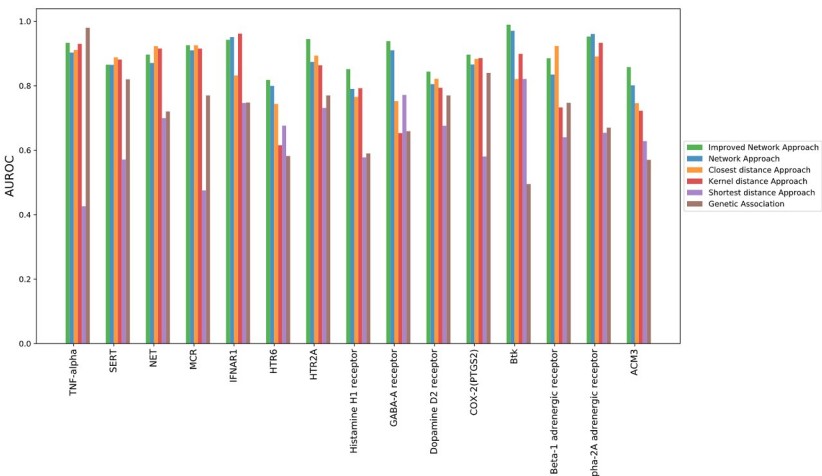

**Fig 2. AUROC performance of network algorithm, improved network algorithm, genetic association and random genes.**

ranking is higher than a given threshold, namely, the ratio of the successfully predicted known drug-indication associations to the total known drug-indication associations. Specificity refers to the percentage of associations that are below the threshold. Receiver-operating characteristics (ROC) curves were plotted by varying the threshold, and the values of area under curves (AUC) were calculated as AUROC for the measurement of the performance of the predictor.

In our performance evaluation results, we found the performance of network approach with the selected drug target always outperformed that with random drug target (Fig 2). The average AUROC of the selected drug targets had reached much higher average AUROC than the random drug targets on predicting the approved indications, which demonstrated the indication predictive ability for the specified drug target doesn't occur by chance.

All the 15 drug targets outperformed random targets by AUC, especially for Alpha-2A adrenergic receptor and GABA-A receptor, which has increased AUC by 0.37 and 0.40, respectively (Figs 3 and 4). The average difference between AUC of specified drug target and random druggable targets is around 0.216. The ROC curves for other targets can be found in S1 and S2 Figs.

## Comparison with other indication prediction approaches

We also compared the performance of our method with other state-of-art approaches. One of the commonly used methods is to identify disease indications by genetic associations between a drug target and specific diseases, e.g., if there is a novel target to disease association was identified from GWAS studies, this disease could be considered as a potential indication for that drug target. In addition to the genetic association approach, we also investigated other methods that were used to predict target indications, such as proximity-based network approaches that proposed by Guney et al [21]. In proximity-based network approaches, 5 different methods were developed to quantify the proximity that defined by network distance between drug targets and diseases. For each drug target to disease pair, the network distance between drug targets and disease genes was calculated by each of the 5 methods. On the other hand, a set of the expected distances that between the same targets with randomly selected disease gene sets were also calculated. The network distance and expected distances were compared to derive the significance of network distance between targets and diseases, on which the diseases can be

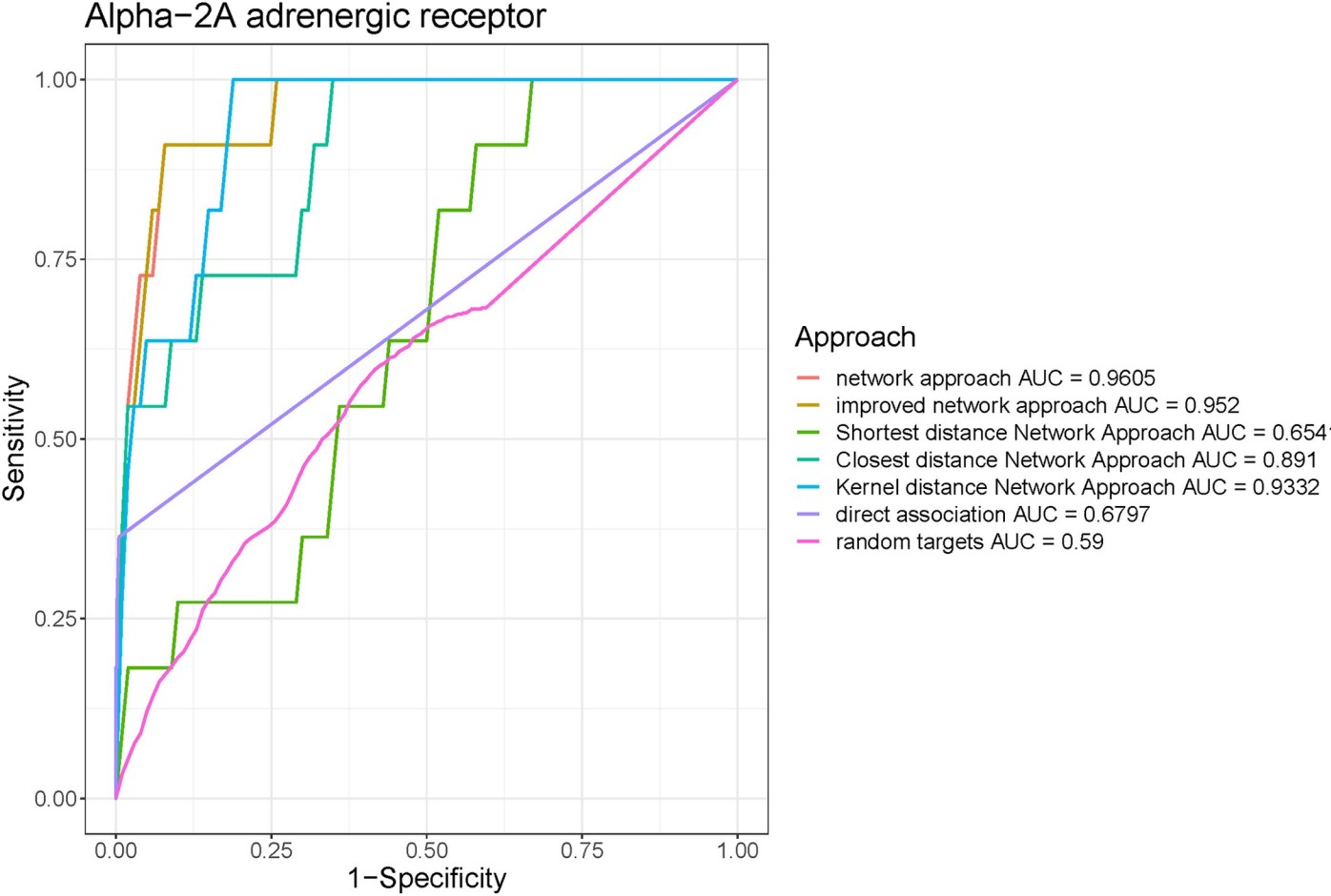

**Fig 3. Receiver Operating Characteristic (ROC) curve showing the performance of different approaches on Alpha-2A adrenergic receptor.** The plot for network approach is characterized by an AUC (area under the curve) of 0.96.

further prioritized for indication prediction. Among the 5 developed the methods, the closest distance approach (averages the shortest path lengths between drug targets to only the nearest disease proteins), the shortest distance (averages the shortest path lengths between drug targets to all the disease proteins), and kernel distance (down-weights the longer shortest path distances using an exponential penalty) demonstrated a better performance in discriminating among the known and unknown drug-disease pairs in author's benchmarking [21]. Therefore, in addition to genetic association approach, we further picked these 3 methods to compare the performance with our method. We compared the performance of the different approaches in the following manner: We derived all genetic associations to diseases for each of the 15 selected drug targets from DisGeNET [20]. Each gene-disease association has been provided with a genetic association score defined by DisGeNET. The score ranges from 0 to 1, indicating the significance of the genetic association. If there are indications that do not associate with any drug targets, the scores will be assigned as 0. For genetic association approach, for each drug target, we sorted the indications by their scores in descending order and used them for indication predictions. For proximity based network approaches, we collected all the genetic associated genes of each disease from DisGeNET, then calculated the network proximities between each drug target and disease genes for indication predictions. For our network approach, we

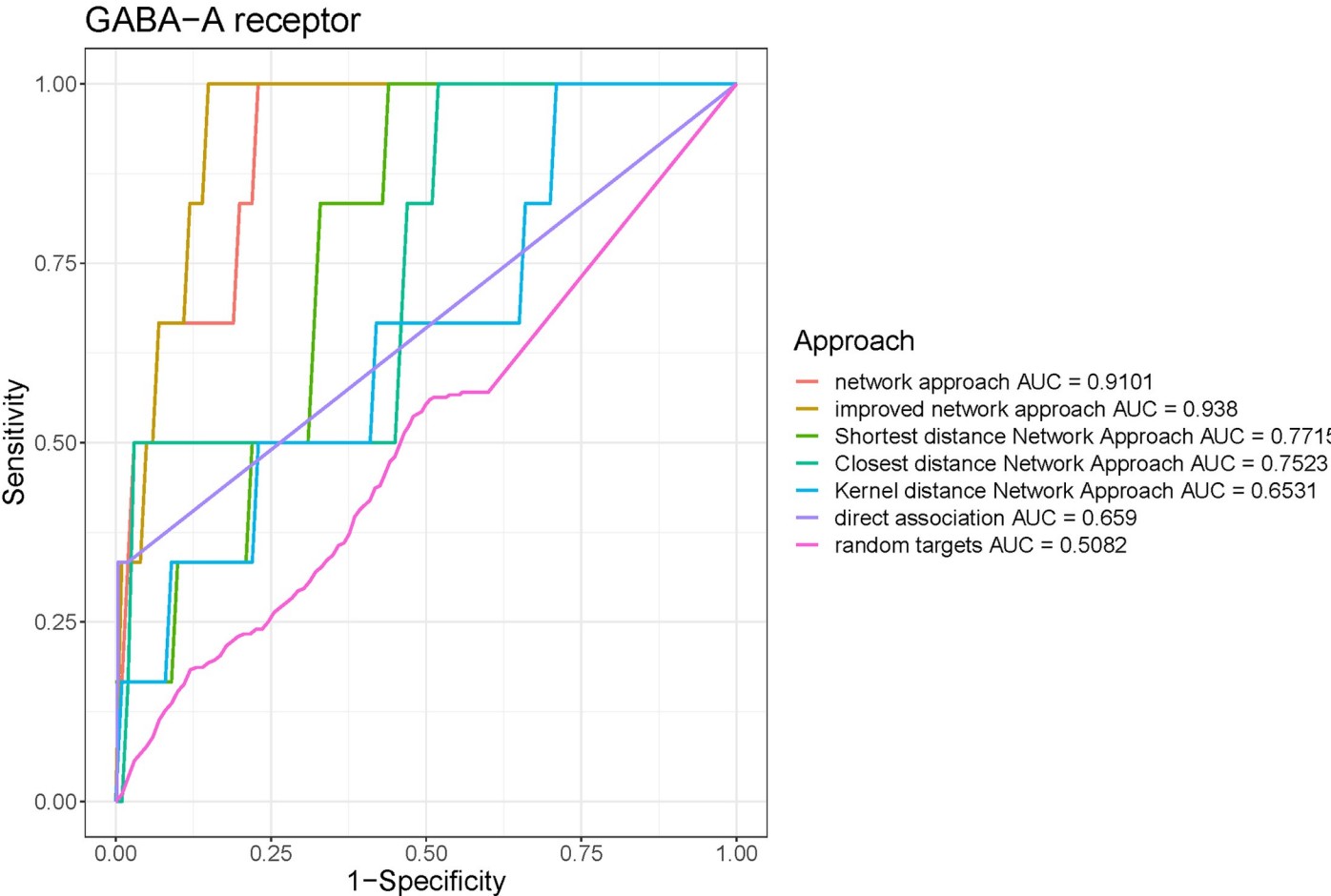

**Fig 4. Receiver Operating Characteristic (ROC) curve showing the performance of different approaches on GABA-A receptor.** The plot for network approach is characterized by an AUC (area under the curve) of 0.91.

used the same disease genes to generate the disease subnetwork for each disease to compare with target subnetwork.

Similarly, we compared the performance of the five methods using AUROC as evaluation metrics. The network-based approach shows superior performance for most drug targets (Fig 2). The average AUC of 15 drug targets from network method is 0.874, which significantly outperformed the average AUC from the genetic association method (0.682). Among the proximity based network approaches, the closet distance approach has an average AUC of 0.848, while the shortest distance approach has an average AUC of 0.645 and 0.833 for Kernel approach. The genetic association approach slightly outperformed the network analysis algorithm for only one target, TNF-alpha.

We also compared the performances with different metrics since under some circumstances, AUROC is not a good metric due to too few indications associated with a specific drug target in DisGeNET. This could potentially prevent predictions for several indications. To address this, we used sensitivity for performance evaluation. We consider sensitivity(also referred as recall) is a good metric since when predicting drug-indication pairs we are most concerned with whether the approved indications are ranked on the top of the list since the true negative control is hard to define. Sensitivity is computed as the proportion of approved

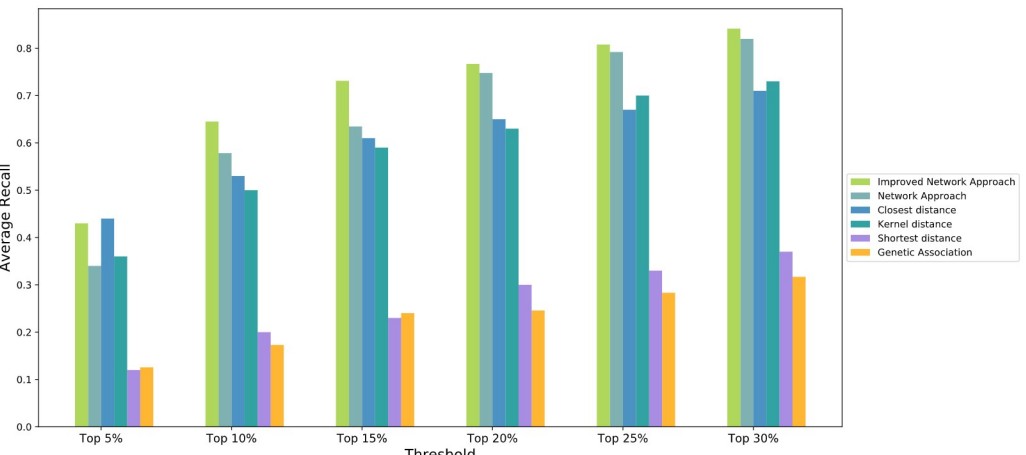

**Fig 5. Sensitivity of predictions on network approach and other approaches.**

indications that are predicted to be positive. Since the number of correctly predicted true positives reflects the discriminatory power of a prediction method to distinguish true positives, especially when the number of negative samples is far larger than that of positive samples. Therefore, we report the number of correctly predicted drug-disease associations with respect to a specified top-rank threshold. A known drug-disease association is considered as correctly predicted if its ranking is higher than a specified top-rank threshold. To compare the performance of the methods by sensitivity, we used different ranking threshold (Top 5%, 10%, 15%, 20%, 25% and 30%) based on the total numbers of diseases as positive, and investigate if the known target disease links can correctly predicted by different methods. among all the. The sensitivity derived from network approach, genetic association approach and proximity based network approaches for the 15 drug targets were averaged and compared in Fig 5.

In Fig 5, it can be seen that our method demonstrated a superior performance than other methods. When the threshold increased to top 10% and above, the average sensitivities were consistently higher than of other methods, indicating our method can correctly predicts more true drug-disease associations upon each top-rank threshold. The shortest distance and genetic association approaches, on the other hand, had the lowest performance, while the closest distance and kernel distance approaches showed good performance as well. Throughout the performance comparison across different approaches, the best threshold on the 15 drug targets for our network method ranges from top 10% to top 20%, the average is around 15%. Therefore, we would recommend top 15% as a threshold to shortlist the predicted candidate indications for further investigation.

## Improved network approach

In the method described in the material and method section on our network approach, equal weights are assigned to all nodes in the disease subnetwork while evaluating target—disease subnetwork node enrichment. However, since disease subnetwork is formed by connecting its genetic associated genes and additional nodes, the genetic associated genes provide much stronger evidence for the association than other additional nodes. Hence, it is sensible to put more weights on genetic associated genes. In our work, we also proposed the improved network approach by putting twice weights on genetic associated genes. Specifically, for each indication, each of its disease associated genes is considered as two nodes in the disease subnetwork. We reperformed hypergeometric test to evaluate how significantly the target

subnetwork node set overlaps with new disease subnetwork node set. Our evaluation result demonstrated that the improved method has reached even higher performance on most of the drug targets when compared to our original network method, the genetic association method and proximity based network methods as well (Figs 2 and 5).

## Case study: Expanding novel indications for highly pursued drug targets

We applied our method on predicting novel indications on several highly pursued drug targets. Here we conduct a case study on a combination of two drug targets: interleukin 12 (IL-12)/IL-23 for their indication expansion to further validate the performance of the proposed method.

The IL-12/IL-23 had attracted immense interest for clinical development, recently. Both IL-12 and IL-23 are important cytokines. Their involved signaling pathway has been identified as a critical role on inducing the inflammation in adaptive immune responses. IL-12 promotes the Th1 polarization and secretion of critical cytokines such as interferon-γ and tumor necrosis factor by T and NK cells [22]. Whilst IL-23 helps to promote the differentiation of naïve T cells into Th17 cells with the secretion of several inflammatory cytokines such as IL-17 and IL-22 [23]. There are various drugs developed to modulate IL-12 (Th1)/IL-23(Th17) pathways, such as IL-23 specific antibodies and IL-23R peptide inhibitors for treating autoinflammatory diseases. On the other hand, combined therapies have also been under immense investigation. Clinical trials have demonstrated the clinical effect with a safety profile on IL-12/IL-23 blockade. Ustekinumab, a therapeutic antibody targeting both cytokines is now widely licensed for the treatment of Crohn's disease [24] as well as psoriasis and psoriatic arthritis [25]. Further studies would elucidate its potential role as first-line therapy for IBD and other autoinflammatory diseases. There is strong interest within biopharma to expand the therapeutic base for Ustekinumab and other similar drugs. Similarly, there is a critical need to explore indication expansion potential for many other therapeutics to benefit patients. This offers an opportunity for leveraging our network approach to systematically explore the drug novel indications that supported by genetic evidences.

To perform the indication expansion study, we generated a target subnetwork using IL-12/IL-23 as drug targets, which contains the two genes and their neighboring interactomes. We also generated disease subnetworks for each indication in DisGeNET based on their genetically associated genes. The network enrichment analysis was performed between the target subnetwork and each disease subnetwork. The computation steps were repeated 3 times, and the enrichment scores were averaged. The diseases were then ranked by the enrichment score.

When we investigated the ranked indications (S2 File), we have found that the approved indications, or indications which are currently under clinical trials for IL-12/IL-23 intervention were highly ranked in our predictions. Among the ranked 6,701 indications from DisGeNET, inflammatory bowel disease (IBD), which is one of the approved indications for Ustekinumab, was ranked within top 30 (14th) predicted indications (Table 2). Other approved indications for Ustekinumab also highly ranked among all the predicted indications: Crohn's disease (165th), Psoriasis (Pustulosis of Palms and Soles, 156th) psoriatic arthritis (359th). Additional IBD related indications such as colitis, was ranked at 70th, and there is a phase III trials in ulcerative colitis [24]. Figs 6 and 7 demonstrated the connections between disease subnetwork and target subnetwork, where the diseases are psoriatic arthritis and hidradenitis suppurativa, respectively. Both networks showed a significant overlap between disease and target subnetwork nodes, indicating a strong relationship between the target perturbed process and the disease functions. The approval and on-going clinical trials of the two indications lend further support to our observations.

**Table 2.  Top30 predicted indications for IL12/IL23.**

| Rank | Disease | P-value | FDR |
|---|---|---|---|
| 1 | Seborrheic dermatitis of scalp | 1.16E-303 | 7.75E-300 |
| 2 | SPINOCEREBELLAR ATAXIA 23 | 2.42E-299 | 1.62E-295 |
| 3 | Sensation Disorders | 2.42E-299 | 1.62E-295 |
| 4 | Plague | 2.42E-299 | 1.62E-295 |
| 5 | Irritable Mood | 2.42E-299 | 1.62E-295 |
| 6 | Onychomycosis | 5.35E-298 | 3.58E-294 |
| 7 | Pentosuria | 3.31E-296 | 2.22E-292 |
| 8 | Bacteroides Infections | 3.31E-296 | 2.22E-292 |
| 9 | Cheilitis | 1.73E-295 | 1.16E-291 |
| 10 | Oral candidiasis | 9.40E-294 | 6.29E-290 |
| 11 | Candidiasis, Chronic Mucocutaneous | 2.28E-290 | 1.52E-286 |
| 12 | Tyrosine Kinase 2 Deficiency | 5.30E-290 | 3.54E-286 |
| 13 | Leukoerythroblastic Anemia | 5.30E-290 | 3.54E-286 |
| 14 | INFLAMMATORY BOWEL DISEASE, AUTOSOMAL RECESSIVE | 5.30E-290 | 3.54E-286 |
| 15 | COLD-INDUCED SWEATING SYNDROME 1 | 2.27E-289 | 1.52E-285 |
| 16 | Bronchial Spasm | 2.27E-289 | 1.52E-285 |
| 17 | Intraabdominal Infections | 3.98E-288 | 2.66E-284 |
| 18 | STUVE-WIEDEMANN SYNDROME | 3.98E-288 | 2.66E-284 |
| 19 | Lobomycosis | 1.06E-287 | 7.06E-284 |
| 20 | Hand, Foot and Mouth Disease | 1.45E-287 | 9.70E-284 |
| 21 | Hyper-Ige Recurrent Infection Syndrome, Autosomal Dominant | 6.21E-287 | 4.15E-283 |
| 22 | Rheumatoid Arthritis, Systemic Juvenile | 2.61E-286 | 1.74E-282 |
| 23 | Gastroenteritis | 3.75E-284 | 2.51E-280 |
| 24 | Amyloidosis, Primary Cutaneous | 6.94E-284 | 4.63E-280 |
| 25 | Entamoeba histolytica Infection | 6.94E-284 | 4.63E-280 |
| 26 | Glutaric aciduria, type 1 | 6.94E-284 | 4.63E-280 |
| 27 | Genital Herpes | 2.08E-283 | 1.39E-279 |
| 28 | Cardiac Output, High | 6.94E-284 | 4.63E-280 |
| 29 | Fetal Resorption | 6.94E-284 | 4.63E-280 |
| 30 | Galactorrhea not associated with childbirth | 6.94E-284 | 4.63E-280 |

For potential novel indications, the first ranked indication is Seborrheic dermatitis of scalp. We found the potential clinical efficacy on dermatitis were supported by multiple studies. In a recent report from Takahashi et al. [26], the authors observed Anti-IL-12/IL-23p40 antibody ameliorates dermatitis and skin barrier dysfunction in mice. Moreover, in a recent phase II, placebo-controlled clinical trial on moderate-to-severe atopic dermatitis patients [27], the ustekinumab group achieved higher clinical responses at the 12, 16 weeks (primary endpoint) and 20 weeks compared to placebo. Although the difference between groups was not significant, the inconclusive results might possibly be due attributed to insufficient dosing of ustekinumab and the unlimited use of background topical corticosteroids. Hidradenitis suppurativa, which was another highly pursued immune mediated indication, was ranked 110th in our predictions. We found that there is one treatment Dimethyl fumarate which impairs IL-12 and IL-23 production by dendritic cells and macrophages, which is currently in clinical phase III trial for this disease [28, 29]. Other top predicted novel indications include Gastroenteritis (23th), which could play a role in the initiation and/or exacerbation of IBD [30].

We also noticed several indications of infectious nature, such as bacteroides infections, candidiasis and intraabdominal infections were among the top ranked indications (Table 2). For

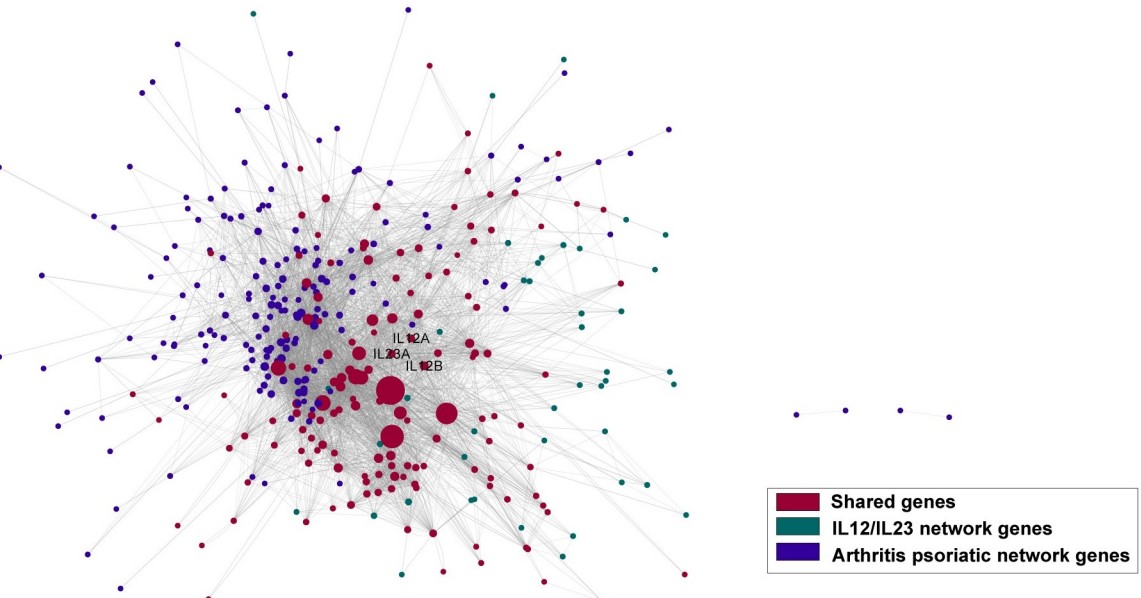

**Fig 6. Target and disease subnetwork connection network.** Purple represents disease (Psoriatic Arthritis) network nodes, green represents IL12/IL23 network nodes, and red represents target and disease subnetwork shared nodes, respectively.

this perspective we found studies demonstrating that psoriasis patients treated with IL-12/23 blockers showed a reduced risk for serious infection compared with those who received TNF or IL-17 inhibitors [31]. This might be due to the restoration of tissue barrier function or normalization of immune dysregulation, suggesting improvement over certain infectious conditions as a result of therapeutic intervention.

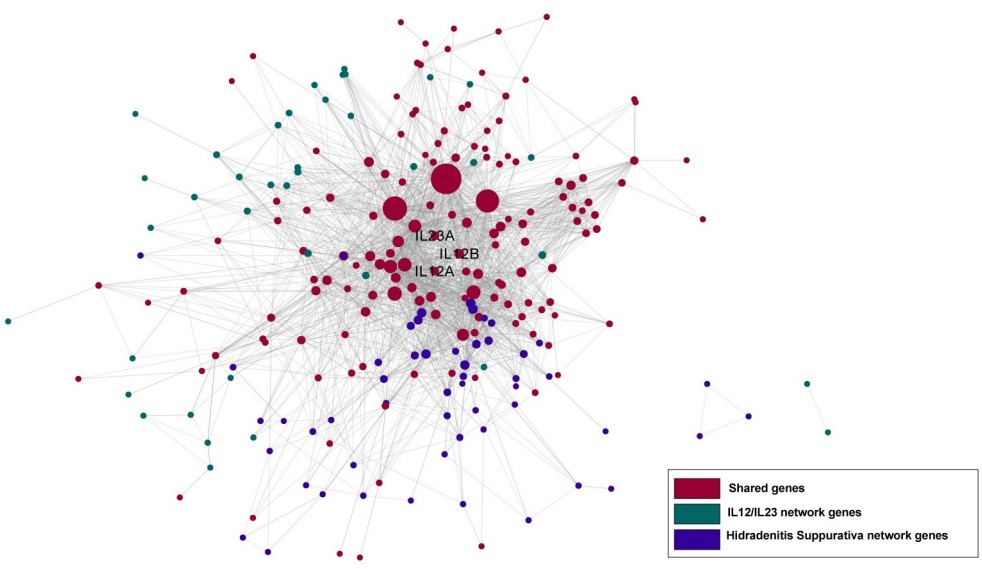

**Fig 7. Target and disease subnetwork connection network.** Purple represents disease (Hidradenitis Suppurativa) network nodes, green represents IL12/IL23 network nodes, and red represents target and disease subnetwork shared nodes, respectively.

There are other novel indications which worth further investigations. They were not listed in Table 2 due to the context limit. The S2 File contains the full list of our predicted indications for IL-12/IL-23. Since our main goal for this work is to develop a method that can systematically prioritize thousands of indications rather than just identify one or two indications for the drug targets, we consider the top rankings on IL12/IL23 from our method could quickly create a shortlist of potential indications (e.g. top 15%) based on the genetic evidence and network connections. This could provide the starting point for bench scientists to systematically evaluate and develop some actionable hypotheses on next steps to act on these potential indication opportunities. They could be further filtered down by other criteria such as biological plausibility, in-house data and literature evidences for the top indications as mentioned above.

## Discussion

In this paper, we proposed a network-based methodology for drug target novel indication expansion by taking advantage of genetic evidence and PPI networks. This methodology successfully recovered FDA approved indications for a set of randomly selected drug targets in our performance evaluation, and it also outperformed other existing indication expansion approaches. The performance was further enhanced by putting weights on genetically associated genes for diseases. On investigating highly pursued drug targets of interest such as IL-12/IL-23, the approved disease indications such as IBD were consistently ranked at the top on our predicted list. We consider the strong performance is due to the genetic evidence identified from genome-wide association studies (GWAS), which led to the identification of multiple variants on IL-12/IL-23 and other neighboring genes associated with IBD, the vast majority of which are shared by Crohn's disease and ulcerative colitis. By further connecting the multiple identified variants and genes into a network module, the disease involved biological processes and pathways can be represented at the cellular level. This will enable us to compare disease associated biological process and pathways that are perturbed by the target engagement. Based on this, several novel indications were also identified based on the consistency and biological plausibility supported by studies on their associations to targets.

We consider this methodology to not only expand potential indications for drug targets but also provide novel insights that help leverage the impact of genetic and pathway information for developing therapeutic intervention strategies. In addition, compared to other indication expansion approaches, our method also has the potential to predict indications for treatments with multiple drug target combinations, other than a single drug target. This is especially useful for recent popular therapeutic approaches, such as combo treatments. Therefore, we consider this methodology to be able to be readily applied to other drug discovery portfolios, which enable us to formulate novel, testable hypotheses facilitating target indication expansion, or drug repositioning candidates, and ultimately realize the goal of personalized medicine.

Despite the encouraging results produced by our methods, some limitations should be noted. First, the methods can only work for diseases with known gene to disease associations, and our current knowledge about gene to disease associations is still far from complete. The predicted results may be biased by heavy reliance on protein interaction topology. It is well known that some diseases originate or impacted by environmental factors, acquired somatic mutations, or other phenomena, which may not be fully characterized by protein interactome network topology, as generated in this workflow. To improve the prediction of target-disease associations for novel indications, other reliable features need to be taken into consideration, such as using tissue-level gene expression to generate disease modules. Therefore, it is important to note that the indication expansion by our approach can only be as accurate as the

current protein interactome can accurately represents the underlying biological process for each disease. Lastly, for some diseases who have known associated genes but if there is no protein interactome data available then we cannot directly use the approach for such cases. An alternative approach would be to consider other types of networks (co-expression or functional networks) and apply the same approaches.

## Materials and methods

### Datasets

We obtained all the available FDA-approved drug targets and their indications from Clarivate's MetaCore for our target indication expansion and performance evaluation. Indications for oncology were excluded as we were not investigating cancer indications. After removing oncology indications, there are 2,806 target-indication pairs within 267 drug targets and 270 diseases in total.

The disease to gene association datasets were downloaded from DisGeNET (http://www. disgenet.org). DisGeNET [20] is one of the public databases which consists of large collections of gene to human disease and phenotype associations. The database collects the information from expert curated repositories, including GWAS catalogues, animal models and scientific literature.

There are several data processing procedures required in order to evaluate whether our ranking list for DisGeNET could better recover the known approved indications of a query target in terms of the area under receiver operating characteristic curve (AUROC) and sensitivity. Firstly, since there is no target to indication approval data from DisGeNET, we needed to obtain the approved indications from other databases and map the approved indications to DisGeNET disease terms. In this work, the approved target to indications list was obtained from MetaCore, including FDA and EU approved indications which were labeled as positive. Secondly, we were not able to match disease terms in DisGeNET and MetaCore directly due to different disease ontologies. Hence, we used Mesh ID as the key to map disease names across two databases. Since DisGeNET collects data from different sources, diseases that are without Mesh ID were removed in our study. Since in DisGeNET there are also many diseases which have the same Mesh ID, to further reduce the redundancy of diseases in our prediction list, we only kept one term for those diseases which ranks highest for each Mesh ID in our result. There are 9, 231 diseases with 6,701 Mesh IDs, therefore we kept 6,701 ranked indications in our IL12/IL23 case study. The full list of the included diseases with their genetic associated genes in our study from DisGeNET can be found in S3 File. We used AUROC as the evaluation metric, which was obtained by comparing ranking list with the ground truth. To further evaluate if approved indications were recovered, we also took FDR < 0.05 as a threshold and calculated sensitivity.

The protein-protein interaction (PPI) network data was downloaded from STRING database (version11) [32]. The STRING database is one of the most comprehensive protein to protein interaction network with predicted and known interactions. Each edge is given a weight to identify the degree of confidence. In order to generate a reliable, high-trust level network reference for our approach, we selected interactions with confidence score greater than 0.7 defined by STRING. After data preprocessing, we reconstructed our global protein-protein interaction network with 14,157 nodes and 326,634 edges.

### Generation of drug target subnetwork

In our work, a drug target subnetwork is defined as a subnetwork that connects the drug target (s) and their neighboring nodes in a PPI network. In order to select the top neighboring nodes

into the subnetwork, a node prioritization algorithm was needed to rank the neighboring nodes according to their connectivity to the target. Many node prioritization algorithms have been developed for this purpose, such as neighborhood scoring [33], Interconnectivity [34], network propagation [35]. Among the algorithms, the network propagation algorithm has the best performance on predicting the disease associated genes in our benchmark. Hence, we selected network propagation to identify the top 200 neighboring nodes for a drug target's subnetwork.

Network Propagation is a flow-based method that prioritizes candidates by smoothing disease-associated information over the network. The scoring of the network nodes can be regarded as propagating flow through the network. The starting nodes of the flow correspond to the drug targets and are assigned a flow of 1, while the remaining network nodes are assigned a flow of 0. These flow assignments represent the prior knowledge of the condition and are smoothed over the network to prioritize candidates that are in close proximity to all disease associated genes. The scoring is done by simulating an iterative process where flow is pumped from the starting nodes to their network neighbors. In addition, every network node propagates the flow received in the previous iteration to its neighbors. The iterations are repeated until a steady state is reached. The final flow that each network object received corresponds to its final score and defines the rank of the object in the list of candidates. In each iteration, the flow for the network objects is updated as follows:

$$F^t = \alpha \cdot A' \cdot F^{t-1} + (1-\alpha) \cdot F^0$$

$F^t$ is a vector containing the flow for each network object at time point $t$. A' corresponds to the adjacency matrix of the graph, where each entry is normalized by the degrees of the source and target nodes. The normalization by node degrees compensates for the fact that nodes with many interactors have a higher chance of picking up flow by chance and are thus more likely to be ranked higher in the prioritization. $F^0$ represents the prior knowledge vector containing the scores for disease genetic associated genes, or differentially expressed genes. The algorithm terminates when the $L_1$ norm of the difference between $F^t$ and $F^{t-1}$ drops below $10^{-6}$.

## Generation of disease subnetworks

We applied the DIAMOnD algorithm [36] to generate disease subnetworks. The algorithm starts from genetically associated genes of a disease as seed node set and adds a new node that is highly connected with the seed nodes iteratively. At every step, DIAMOnD pre-selects a set of candidate nodes based on their connectivity to the current seed node set. The overconnectivity p-value for each candidate node is calculated using hypergeometric test. Under null hypothesis, the intersection size between a node neighbors and seed nodes is a random variable following hypergeometric distribution.

The node with the smallest p value will be added into seed set each time. The growth stops at a user-defined threshold, yielding one or more start nodes-enriched subnetworks. The number of iterations to add candidate nodes is assigned manually in the original DIAMOnD paper. The author chose 200 as the putative size of complete disease modules within the interactome for evaluating the performance of DIAMOnD. To further explore the effect of network size parameter on performance, we did a benchmarking on different node numbers in DIAMOnD as 100, 150, 200, 250 and 300 respectively. The benchmarking was conducted on the 15 selected drug targets and used their approved indications as ground truth to evaluate ranking performance. As a result of the benchmarking, we didn't identify any significant difference among the ranking performance across these 5 parameters. The performance on average AUROC and recall of 15 drug targets in different subnetwork size is shown in Table 3.

**Table 3. Ranking performance based on different number of nodes added to disease subnetwork.**

| Network nodes | n = 100 | n = 150 | n = 200 | n = 250 | n = 300 |
|---|---|---|---|---|---|
| Avg AUROC | 0.8737 | 0.8828 | 0.8741 | 0.8740 | 0.8705 |
| Avg Recall | 0.5847 | 0.6340 | 0.6793 | 0.6873 | 0.676 |

Therefore in our calculation, we set the iteration number as 200. The original seed node set which are the disease associated genes and the additional added nodes by the iteration were used to generate disease subnetwork.

## Comparison between target-disease subnetworks

We applied hypergeometric test to compare the two subnetworks and obtained p value to evaluate the topological enrichment. To account for multiple tests, false discovery rate (FDR) is generated by adjusting p value with the correction method introduced by Holm et al [37]. Furthermore, as the genetic associated genes vary from disease to disease, we assumed the disease subnetwork started with more associated genes are much more reliable. Then the metrics is adjusted by number of associated genes with p value, N refers to number of associated genes of the disease:

$$Metric = \begin{cases} \dfrac{P\ val}{log_2(N+1)}, & P\ val < 1 \\ 1, & P\ val = 1 \end{cases}$$

Diseases were sorted by the metric in an ascending order, the inverse of rank was calculated as the final association score.

## Supporting information

**S1 Fig. ROC curves for randomly selected targets showing the performance of different approaches.**
(TIF)

**S2 Fig. ROC curves for randomly selected targets showing the performance of different approaches.**
(TIF)

**S1 File. Full list of all approved indications for the 15 randomly selected drug targets.**
(XLSX)

**S2 File. Full list of predicted indications for IL-12/IL-23.**
(CSV)

**S3 File. Full list of diseases with genetic associated genes included in this study from Dis-GeNET.**
(CSV)

## Author Contributions

**Conceptualization:** Cheng Zhu.

**Data curation:** Yingnan Han.

**Formal analysis:** Yingnan Han, Cheng Zhu.

**Methodology:** Yingnan Han, Cheng Zhu.

**Project administration:** Deepak K. Rajpal.

**Resources:** Clarence Wang, Katherine Klinger, Deepak K. Rajpal, Cheng Zhu.

**Supervision:** Clarence Wang, Katherine Klinger, Deepak K. Rajpal, Cheng Zhu.

**Validation:** Yingnan Han.

**Visualization:** Yingnan Han.

**Writing – original draft:** Yingnan Han, Cheng Zhu.

**Writing – review & editing:** Katherine Klinger, Deepak K. Rajpal, Cheng Zhu.

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
