## [Decision Letter · Decision Letter 0]

3 Mar 2021

PONE-D-21-01587

An integrative network-based approach for drug target indication expansion

PLOS ONE

Dear Dr. Zhu,

Thank you for submitting your manuscript to PLOS ONE. After careful consideration, we feel that it has merit but does not fully meet PLOS ONE’s publication criteria as it currently stands. Therefore, we invite you to submit a revised version of the manuscript that addresses the points raised during the review process.

Both reviewers raised a number of technical concerns which should be individually addressed in a revised manuscript. While the authors may choose to highlight novel aspects of their approach, it is not essential to address comments regarding novelty as this is not a required criterion for publication in PLoS One.

We look forward to receiving your revised manuscript.

Kind regards,

Jishnu Das, Ph.D.

Academic Editor

PLOS ONE

2. Thank you for stating the following in the Competing Interests:

We note that one or more of the authors have an affiliation to the commercial funders of this research study : Sanofi.

(2) Please also provide an updated Competing Interests Statement declaring this commercial affiliation along with any other relevant declarations relating to employment, consultancy, patents, products in development, or marketed products, etc.  

Reviewers' comments:

Reviewer's Responses to Questions

**Comments to the Author**

1. Is the manuscript technically sound, and do the data support the conclusions?

Reviewer #1: Partly

Reviewer #2: Partly

2. Has the statistical analysis been performed appropriately and rigorously? 

Reviewer #1: Yes

Reviewer #2: No

3. Have the authors made all data underlying the findings in their manuscript fully available?

Reviewer #1: Yes

Reviewer #2: Yes

4. Is the manuscript presented in an intelligible fashion and written in standard English?

Reviewer #1: Yes

Reviewer #2: Yes

5. Review Comments to the Author

Reviewer #1: In this manuscript, Zhu et al. propose a network-based approach for predicting new drug target-indication pairs. Their method integrates information from protein-protein interaction networks and disease-gene associations and provides insights into the roles of biological pathways for drug repurposing. Their method is novel, easy to implement, and is presented in a clear fashion. Nevertheless, there are a few concerns that need addressing before I can recommend the manuscript for publication.

1. The authors seem to have constructed drug target subnetworks and disease subnetworks in different ways. I am curious about the performance if the authors generate the disease subnetwork in the same way as they generate the target subnetwork, i.e. initializing disease genes as a flow of 1 and other nodes as a flow of 0 and performing network propagation. What is the advantage of using the DIAMOnD algorithm for generating the disease subnetwork?

2. In addition to protein-protein interaction networks, I wonder if the method could be improved by integrating other types of networks connecting genes, e.g. genetic interaction networks. Ideally, genetic interaction networks would give a better representation of pathway associations between genes.

3. By using newly predicted target-indication associations, the method can essentially perform drug-indication prediction by connecting drugs with associations predicted for their known targets. How does that compare to existing drug-indication prediction methods? The authors should compare against a few state-of-the-art drug-indication prediction methods to show the superiority and usefulness of their method.

Minor comment:

In line 415-417 on page 21, should the drug target of interest be initialized with a flow of 1, instead of “disease associated genes”?

Reviewer #2: The authors introduce a network-based method for predicting novel indications for existing drugs. Starting from the drug's target, they generate a network of neighboring genes based on known protein-protein interactions. They compare this network to similarly constructed networks of genes associated with clinical indications and flag those indications that have high overlap with the drug target's network. The method uncovers known indications of 15 drugs and predicts novel indications for a drug that targets IL12/IL23. The authors support these results with literature linking the targets to the new indications.

Given the cost and effort of developing new drugs, methods for repurposing approved drugs for additional indications are in high demand and the subject of the paper is relevant and interesting. Nevertheless, the manuscript has three deficiencies. First, it is not clear that the proposed method is novel. Second, the validation is not performed correctly. Third, the utility of the method for practical drug repurposing appears low. These aspects are addressed individually below.

Network-based methods for associating drugs, targets and diseases have been around for at least a decade. Barabasi's 2011 Nature Reviews Genetics piece, which the authors cite (Ref 17), discusses network-based pharmacology and provides numerous references to similar work. What aspects of the current work set it apart from the literature? This manuscript applies standard methods of graph theory and network-based inference, and the authors should make clear exactly how the current method differs from its predecessors, both in its methods and its results.

To properly validate the method, the authors should compare it to existing methods and demonstrate that it performs at least as well. Ideally, they will also provide evidence that the method generates novel predictions. The validation, as presented, does not establish the power of this method. In the first validation, the authors compare their method's ability to identify indications for 15 targets with the performance of the same method starting from random targets. The purpose of comparing the method to itself, it seems, is to demonstrate that it generates "meaningful" results. However, using AUROC for comparing the method starting from a known target to the method starting from a random target is problematic. The purpose of the method is to find new indications for existing targets, which implies that current target-indication information is incomplete. How can one then discern between "correct" and "incorrect" links between targets and indications? As new target-indication links are discovered, the results of the analysis will change. The validation method requires full knowledge of links between targets and indications, yet the purpose of the method is to uncover as-yet unknown links.

In the test case of IL12/IL23, the authors predict thousands of potential indications and then cherry-pick ones that are linked to the cytokines through literature. Although this gives some support to the method, it doesn't show that existing methods are incapable of identifying the same conditions, and it doesn't show that randomly selected indications will not have literature support. How many novel indications for IL12/IL23 should one expect to find by chance? What if one were to use an alternative network method? It's good that the authors compare to gene association, but they should measure their performance against multiple competing techniques.

The method does not seem useful for selecting indications for repurposing. It finds nearly 2600 indications that relate to IL12/IL23 with p-values less than 1e-20. How does one choose which indications to pursue? The time and effort required to experimentally test whether a drug can be repurposed in each of the 2600 indications is prohibitive. If the purpose is to identify potential uses for existing drugs, then it will be necessary to further limit the number of indications predicted for a target. Can the authors provide guidance on how to prioritize indications for repurposing?

At minimum, the paper needs a more thorough analysis of the proposed method and its features. For example, target subnetworks and disease gene subnetworks have 200 nodes each in the method. How does this arbitrary value affect the prediction? It seems odd that all diseases have exactly 200 genes associated with them. Wouldn't a p-value cutoff for DIAMOnD be better? The manuscript would benefit from an analysis of how each of the parameters used in the method affect its performance.

General points:

It's not clear how the "improved" method differs from the original method. Can the authors explain this and provide enough information for the two methods to be independently replicated?

File S3 contains almost 300k lines, most of which are duplicates. Why is this? It's an inconvenient format for fellow researchers.

In Methods, the section "Generation of drug target subnetwork" starts by addressing drug targets, and switches to networks of disease associated genes. The description of Network Propagation algorithm refers to disease associated genes. Which is it, drug targets or disease associated genes?

6. PLOS authors have the option to publish the peer review history of their article (what does this mean?). If published, this will include your full peer review and any attached files.

Reviewer #1: No

Reviewer #2: **Yes: **Timothy R. Lezon

---

## [Author Response · Author response to Decision Letter 0]

3 May 2021

Our response to reviewer’s comments were provided as follows, as well as in our response letter to reviewers. The updated Funding Statement and Competing Interests Statement were also included in our cover letter for the revised manuscript.

Reviewer #1: In this manuscript, Zhu et al. propose a network-based approach for predicting new drug target-indication pairs. Their method integrates information from protein-protein interaction networks and disease-gene associations and provides insights into the roles of biological pathways for drug repurposing. Their method is novel, easy to implement, and is presented in a clear fashion. Nevertheless, there are a few concerns that need addressing before I can recommend the manuscript for publication.

1. The authors seem to have constructed drug target subnetworks and disease subnetworks in different ways. I am curious about the performance if the authors generate the disease subnetwork in the same way as they generate the target subnetwork, i.e. initializing disease genes as a flow of 1 and other nodes as a flow of 0 and performing network propagation. What is the advantage of using the DIAMOnD algorithm for generating the disease subnetwork?

Response: We are grateful for providing us an opportunity to explain the rationale for utilizing DIAMOnD. For the current study, we believed that DIAMonD would be appropriate for generating disease subnetwork : 

The network propagation algorithm started from one (or a few) starting nodes which generate flows iteratively that spread to the whole network, and other nodes in the network receive flows from the starting nodes and ranked by the flows that they received. The neighboring nodes of starting nodes will likely be ranked higher than other nodes as they usually receive more flows. For generating a target subnetwork, it is straightforward to generate a well-connected target subnetwork which starts from only one or two drug targets, with top ranked neighboring nodes added to represent perturbed biological process by the target. 

For disease subnetwork, since a disease usually has multiple associated genes, it is more of a challenge to build a subnetwork that can well connect all disease genes into a subnetwork. Using network propagation to generate disease subnetwork will result in many sub-clusters in the subnetwork which may fail to best represent the disease dysregulated processes. The DIAMOnD algorithm appeared to address this challenge in a cohesive manner as it defines a disease’s genes as seed node set, then at each step, it iteratively selects a best candidate from other nodes which builds the best connectivity to the seed node set until seed node set are well connected. Therefore, in employing DIAMOnD we realized the maximum connectivity among disease genes and the disease dysregulated processes are represented comprehensively.

2. In addition to protein-protein interaction networks, I wonder if the method could be improved by integrating other types of networks connecting genes, e.g. genetic interaction networks. Ideally, genetic interaction networks would give a better representation of pathway associations between genes.

Response: We are extremely thankful for this suggestion. We reasoned that since PPI network was well accepted and widely used in multiple published network studies, it would be good to establish our work with this approach. We do agree that integrating other types of network approaches may have the potential to improve the performance, as we covered this during the course of our discussion section of the manuscript. Since the current scope of our work is focused on the proposal of a novel network method to systematically prioritize target indications with benchmarking and comparing to other state-of-art algorithms in a PPI network. We really like the suggestion, would definitely conduct further studies to improve the method performance by integrating additional network data when we expand the scope of our work on these approaches in future. 

3. By using newly predicted target-indication associations, the method can essentially perform drug-indication prediction by connecting drugs with associations predicted for their known targets. How does that compare to existing drug-indication prediction methods? The authors should compare against a few state-of-the-art drug-indication prediction methods to show the superiority and usefulness of their method.

Response: A very nice suggestion by the reviewer. As suggested, in addition to the direct genetic association approach, we further compared our method with proximity based network approaches (Emre Guney et al. https://www.nature.com/articles/ncomms10331)developed from Barábasi lab. The proximity-based network based approaches include Closest, shortest, Kernel based methods that quantify the network-based relationship between drug target and disease proteins, in order to rank the diseases to for target indication exploration. We compared our approach with proximity-based approaches by predicting the approved indications for selected drug targets. The comparison demonstrated that our prediction performance is higher those methods. We have included these details in the revised manuscript starting at line 202. We believe that this suggestion further strengthened the manuscript and we are extremely thankful to the reviewer for this. 

Minor comment:

In line 415-417 on page 21, should the drug target of interest be initialized with a flow of 1, instead of “disease associated genes”?

Response: Thank you for this comment, and you will find that the correction in the manuscript now reads: “The starting nodes of the flow correspond to the drug targets and are assigned a flow of 1”. 

Reviewer #2: The authors introduce a network-based method for predicting novel indications for existing drugs. Starting from the drug's target, they generate a network of neighboring genes based on known protein-protein interactions. They compare this network to similarly constructed networks of genes associated with clinical indications and flag those indications that have high overlap with the drug target's network. The method uncovers known indications of 15 drugs and predicts novel indications for a drug that targets IL12/IL23. The authors support these results with literature linking the targets to the new indications.

Given the cost and effort of developing new drugs, methods for repurposing approved drugs for additional indications are in high demand and the subject of the paper is relevant and interesting. Nevertheless, the manuscript has three deficiencies. First, it is not clear that the proposed method is novel. Second, the validation is not performed correctly. Third, the utility of the method for practical drug repurposing appears low. These aspects are addressed individually below.

Network-based methods for associating drugs, targets and diseases have been around for at least a decade. Barabasi's 2011 Nature Reviews Genetics piece, which the authors cite (Ref 17), discusses network-based pharmacology and provides numerous references to similar work. What aspects of the current work set it apart from the literature? This manuscript applies standard methods of graph theory and network-based inference, and the authors should make clear exactly how the current method differs from its predecessors, both in its methods and its results.

Response: We are extremely thankful to the reviewer for these comments. To address this, we would like to suggest that the discussion of previous reference work was addressed in the introduction part, where we’ve discussed multiple predecessors and their limitations. In our method, we took advantage of the ever increasing human genetics data on human diseases. We hypothesized that for any disease indication, when the genetically associated gene network overlaps with the drug target network that is derived from a PPI network, they could be considered as potential candidates for the drug target. To verify this hypothesis, we performed systematic benchmarking and comparisons with other approaches based on DisGeNet data, including the proximity-based network approaches (Emre Guney et al., https://www.nature.com/articles/ncomms10331) from Barabasi lab. The benchmarking and comparison results showed our approach demonstrated better performance. 

To properly validate the method, the authors should compare it to existing methods and demonstrate that it performs at least as well. Ideally, they will also provide evidence that the method generates novel predictions. The validation, as presented, does not establish the power of this method. In the first validation, the authors compare their method's ability to identify indications for 15 targets with the performance of the same method starting from random targets. The purpose of comparing the method to itself, it seems, is to demonstrate that it generates "meaningful" results. However, using AUROC for comparing the method starting from a known target to the method starting from a random target is problematic. The purpose of the method is to find new indications for existing targets, which implies that current target-indication information is incomplete. How can one then discern between "correct" and "incorrect" links between targets and indications? As new target-indication links are discovered, the results of the analysis will change. The validation method requires full knowledge of links between targets and indications, yet the purpose of the method is to uncover as-yet unknown links.

Response: We thank the reviewer for the suggestions. As suggested, for comparison with existing network methods, in addition to the direct genetic association approach, we further compared our method to several other network methods such as proximity-based network approaches that developed from Barábasi lab. By recovering the approved indications for known drug targets, the comparison demonstrated that our prediction performance is superior. We have provided these details in the revised manuscript in the “Comparison with other indication prediction approaches” section. 

For performance comparison with random targets, the goal of this validation exercise is to verify the top predictions from our method weren’t just by chance. To do this, we defined the known indications of “right” target as expected positive links. We respectively made predictions using the “right” target as well as several random targets and checked if the known indications are ranked high for “right” target and low for random targets.

Because the goal in our study is to see whether the known indications of “right” target are predicted on the top than random targets, we used AUROC as a metric to show where the known indications are in the predictions. As suggested any metrics that evaluate ranking performance could potentially be employed here. 

We agree the exact AUROC may be changed as new target-indication links are discovered. However, we do believe that the conclusions made from the comparison results should still hold true.

In the test case of IL12/IL23, the authors predict thousands of potential indications and then cherry-pick ones that are linked to the cytokines through literature. Although this gives some support to the method, it doesn't show that existing methods are incapable of identifying the same conditions, and it doesn't show that randomly selected indications will not have literature support. How many novel indications for IL12/IL23 should one expect to find by chance? What if one were to use an alternative network method? It's good that the authors compare to gene association, but they should measure their performance against multiple competing techniques.

Response: This is a great comment with respect to indication prioritization efforts in drug discovery. Our main goal for this work is to systematically prioritize thousands of indications, rather than just predict one or two indications for drug targets. The top rankings from our method could quickly create a shortlist of potential indications (e.g. top 15%) based on the genetic evidence and network connections. This could provide the starting point for bench scientists to systematically evaluate and develop some actionable hypotheses on next steps to act on these potential indication opportunities. They could be further filtered down by other criteria such as biological plausibility, literature evidence and in-house data. We used literature evidence as another source of evidence to narrow down the shortlist further rather than cherry-pick to support our predictions. The statements were also added to the IL12/IL23 use case section to clarify this. 

We do agree that there are many other existing alternative methods. Our study demonstrates throughout our benchmarking and comparison with other methods, we showed that our method has better performance on recovering the approved indications (in terms of AUROC and sensitivity). That is to say, the promising indications will have a higher chance in our top rankings than other methods, and this will give scientist better chance and confidence to identify the right indications through thousands of candidates. 

The method does not seem useful for selecting indications for repurposing. It finds nearly 2600 indications that relate to IL12/IL23 with p-values less than 1e-20. How does one choose which indications to pursue? The time and effort required to experimentally test whether a drug can be repurposed in each of the 2600 indications is prohibitive. If the purpose is to identify potential uses for existing drugs, then it will be necessary to further limit the number of indications predicted for a target. Can the authors provide guidance on how to prioritize indications for repurposing?

Response: We can definitely understand the view of the reviewer, and clearly agree that higher number of predictions could prove challenging. We proposed that in our method, p-value is an indicator showing the significance of overlap between drug-target subnetwork and disease subnetwork, and prudent use of it would aid in indication selection. On the other hand, the disease ranking, and percentile are also important indicators for selecting candidate indications. According to the ROC curve of 15 drug targets, we can select the best threshold for a shortlist. We found the best threshold among the 15 drug targets range from top 10% to top 20%, the average is around 15%. We would recommend top 15% as a threshold to shortlist the candidate indications for further investigation.

At minimum, the paper needs a more thorough analysis of the proposed method and its features. For example, target subnetworks and disease gene subnetworks have 200 nodes each in the method. How does this arbitrary value affect the prediction? It seems odd that all diseases have exactly 200 genes associated with them. Wouldn't a p-value cutoff for DIAMOnD be better? The manuscript would benefit from an analysis of how each of the parameters used in the method affect its performance.

Response: We thank the reviewer for these suggestions. The DIAMOnD algorithm iteratively added new candidate nodes that can best connect all the disease associated genes to form a disease subnetwork, e.g. add 200 new genes in addition to disease associated genes. Since we didn’t mean that each disease has exactly 200 associated genes, based on the suggestions by the reviewer, the manuscript has been revised to avoid such confusion and provide more clarity at line 494: “Therefore in our calculation, we set the iteration number as 200. The original seed node set which are the disease associated genes and the additional added nodes by the iteration were used to generate disease subnetwork.” The default setting for newly add nodes in our method is 200, since in the original DIAMOnD paper, author chose 200 as the putative size of complete disease modules within the interactome for evaluating the performance of DIAMOnD. To further address the analysis on network size parameter with performance, we did a benchmarking on adding different node numbers in DIAMOnD as 100, 150, 200, 250 and 300 respectively. The benchmarking was conducted on the 15 selected Drug targets and used their approved indications as ground truth to evaluate ranking performance. As a result of the benchmarking, we didn’t identify any significant difference among the ranking performance of these 5 parameters. We have included more details in the “Generation of disease subnetworks” section.

General points:

It's not clear how the "improved" method differs from the original method. Can the authors explain this and provide enough information for the two methods to be independently replicated?

Response: While the two approaches appear similar, the key difference lies in the fact that the improved method puts more weights on disease associated genes when comparing the overlap between disease subnetwork nodes and drug subnetwork nodes through hypergeometric test. For the network comparison in improved method, each disease associated genes will be considered as two identical nodes in disease subnetwork node set. If there is any disease associated gene overlapped with a gene from target subnetwork node set, it counts this node overlap twice which may lead to a more significant p-value and higher disease rank, so that the prediction performance could be further improved.

File S3 contains almost 300k lines, most of which are duplicates. Why is this? It's an inconvenient format for fellow researchers.

Response: We can understand the way this would have come across. The File S3 is the gene-disease association data from DisGeNET that we used for extracting the disease-associated genes and generating the disease subnetwork. DisGeNET is one of the largest publicly available platforms that provide collections of gene and variant associations to human diseases. The collections were from different genetic data resources, such as HPO, BEFREE and Orphanet. Since each row is a unique disease-gene association, there are no real duplicates and are from different resources as mentioned above. We attached the full file here and make the data available for fellow researchers.

In Methods, the section "Generation of drug target subnetwork" starts by addressing drug targets, and switches to networks of disease associated genes. The description of Network Propagation algorithm refers to disease associated genes. Which is it, drug targets or disease associated genes?

Response: We thank the reviewers for pointing this out. The starting nodes of Network Propagation algorithm in our study refers to drug target, we corrected this in our revised manuscript as “The starting nodes of the flow correspond to the drug targets and are assigned a flow of 1”.

---

## [Decision Letter · Decision Letter 1]

9 Jun 2021

An integrative network-based approach for drug target indication expansion

PONE-D-21-01587R1

Dear Dr. Zhu,

We’re pleased to inform you that your manuscript has been judged scientifically suitable for publication and will be formally accepted for publication once it meets all outstanding technical requirements.

Kind regards,

Jishnu Das, Ph.D.

Academic Editor

PLOS ONE

Additional Editor Comments (optional):

Reviewers' comments:

Reviewer's Responses to Questions

**Comments to the Author**

1. If the authors have adequately addressed your comments raised in a previous round of review and you feel that this manuscript is now acceptable for publication, you may indicate that here to bypass the “Comments to the Author” section, enter your conflict of interest statement in the “Confidential to Editor” section, and submit your "Accept" recommendation.

Reviewer #2: All comments have been addressed

2. Is the manuscript technically sound, and do the data support the conclusions?

Reviewer #2: Yes

3. Has the statistical analysis been performed appropriately and rigorously? 

Reviewer #2: Yes

4. Have the authors made all data underlying the findings in their manuscript fully available?

Reviewer #2: Yes

5. Is the manuscript presented in an intelligible fashion and written in standard English?

Reviewer #2: Yes

6. Review Comments to the Author

Reviewer #2: (No Response)

7. PLOS authors have the option to publish the peer review history of their article (what does this mean?). If published, this will include your full peer review and any attached files.

Reviewer #2: **Yes: **Timothy R. Lezon

---

## [Editor Report · Acceptance letter]

1 Jul 2021

PONE-D-21-01587R1 

An integrative network-based approach for drug target indication expansion 

Dear Dr. Zhu:

I'm pleased to inform you that your manuscript has been deemed suitable for publication in PLOS ONE. Congratulations! Your manuscript is now with our production department. 

Kind regards, 

on behalf of

Dr. Jishnu Das 

Academic Editor

PLOS ONE